# Identification and Characterization of a Novel Species of Genus *Akkermansia* with Metabolic Health Effects in a Diet-Induced Obesity Mouse Model

**DOI:** 10.3390/cells11132084

**Published:** 2022-06-30

**Authors:** Ritesh Kumar, Helene Kane, Qiong Wang, Ashley Hibberd, Henrik Max Jensen, Hye-Sook Kim, Steffen Yde Bak, Isabelle Auzanneau, Stéphanie Bry, Niels Christensen, Andrew Friedman, Pia Rasinkangas, Arthur C. Ouwehand, Sofia D. Forssten, Oliver Hasselwander

**Affiliations:** 1Health & Biosciences, International Flavors & Fragrances, Inc. (IFF), Wilmington, DE 19803, USA; helene-m.a.kane@iff.com (H.K.); qiong.wang@iff.com (Q.W.); hskim1214@gmail.com (H.-S.K.); andrew.friedman@iff.com (A.F.); 2Health & Biosciences, IFF, Saint Louis, MO 63110, USA; ashley.hibberd@iff.com; 3Health & Biosciences, IFF, 8220 Brabrand, Denmark; henrik.max.jensen@iff.com (H.M.J.); steffen.yde.bak@iff.com (S.Y.B.); niels.christensen@iff.com (N.C.); 4Health & Biosciences, IFF, 86270 Dange, France; isabelle.auzanneau@iff.com (I.A.); stephanie.bry@iff.com (S.B.); 5Health & Biosciences, IFF, 02460 Kantvik, Finland; pia.rasinkangas@iff.com (P.R.); arthur.ouwehand@iff.com (A.C.O.); sofia.forssten@iff.com (S.D.F.); 6Health & Biosciences, IFF, c/o Danisco UK Ltd., Reigate RH2 9PW, UK; oliver.hasselwander@iff.com

**Keywords:** microbiome, *Akkermansia* sp., metabolic health

## Abstract

*Akkermansia muciniphila* is a well-known bacterium with the ability to degrade mucin. This metabolic capability is believed to play an important role in the colonization of this bacterium in the gut. In this study, we report the identification and characterization of a novel *Akkermansia* sp. DSM 33459 isolated from human feces of a healthy donor. Phylogenetic analysis based on the genome-wide average nucleotide identity indicated that the *Akkermansia* sp. DSM 33459 has only 87.5% similarity with the type strain *A. muciniphila* ATCC BAA-835. *Akkermansia* sp. DSM 33459 showed significant differences in its fatty acid profile and carbon utilization as compared to the type strain. The *Akkermansia* sp. DSM 33459 strain was tested in a preclinical obesity model to determine its effect on metabolic markers. *Akkermansia* sp. DSM 33459 showed significant improvement in body weight, total fat weight, and resistin and insulin levels. Interestingly, these effects were more pronounced with the live form as compared to a pasteurized form of the strain. The strain showed production of agmatine, suggesting a potential novel mechanism for supporting metabolic and cognitive health. Based on its phenotypic features and phylogenetic position, it is proposed that this isolate represents a novel species in the genus *Akkermansia* and a promising therapeutic candidate for the management of metabolic diseases.

## 1. Introduction

The genus *Akkermansia* was first described by Derrien et al., in 2004 while studying mucin-degrading bacteria present in the human intestine [1]. An isolate obtained from a fecal sample of a healthy human subject was proposed as type strain MucT (=ATCC BAA-835) of the novel species *Akkermansia muciniphila* [1]. Since then, a second species of the *Akkermansia* genus has been proposed by Ouwerkerk et al., in 2016, based on an isolate from reticulated python feces [2]. 

The human gut harbors thousands of bacterial species, which play a vital role in host health and physiology. *A. muciniphila* is commonly present in the human gastrointestinal (GI) tract [3] and is the only known species of the Verrucomicrobia phylum in the mammalian gut [4]. Pangenomic studies of *A. muciniphila* have identified four distinct phylogroups (clades AmI to IV) [5,6] and genotypic and phenotypic diversity among human isolates of *A. muciniphila* were described by Becken et al., 2021 [4].

Despite differences in GI tract anatomy, diet, mucin types, and composition, *A. muciniphila* has been detected in different animals [7]. The capability of *A. muciniphila* in mucin degradation and utilization makes it a key organism at the mucosal interface in the gut. It plays an important role in the expression, production, and recycling of the mucin layer. A small proportion of gut microbiota can degrade host mucin and this metabolic capability provides nutrients for other resident bacteria through cross feeding [8]. The use of mucin as a nutrient source gives *A. muciniphila* an advantage over other bacterial species when colonizing the host, as it can derive nutrients from mucin and may not be dependent on the food intake of the host.

Several studies show an association between *A. muciniphila* and beneficial impacts on host health and physiology [5,9,10,11]. In addition, several animal and clinical studies have shown that *Akkermansia* abundance is significantly reduced in subjects with a compromised metabolic state [12,13,14,15]. These studies highlight the ability of *Akkermansia* to interact with the host and the other resident microbiota to enhance the intestinal barrier integrity and promote an anti-inflammatory state. Recently, a proof-of-concept study accessed on 1 February 2022 (ClinicalTrials.gov Identifier: NCT02637115), showed that after three months of supplementation, *A. muciniphila* consumption was safe and well tolerated, improved several metabolic parameters, and reduced levels of the relevant blood markers for liver dysfunction and inflammation, while the overall gut microbiome structure was unaffected [16]. In addition, a human intervention study (ClinicalTrials.gov Identifier: NCT03893422) in subjects with type 2 diabetes receiving metformin treatments showed that a mixture of butyrate-producing commensal bacteria and an *A. muciniphila* strain improved glucose control after 12 weeks of supplementation [17].

This report highlights the isolation and characterization of a new species of *Akkermansia* from human feces of a healthy, lean subject. *Akkermansia* sp. DSM 33459 showed marked differences in its fatty acid profile and carbon utilization when compared to the type strain *A. muciniphila* ATCC BAA-835. *Akkermansia* sp. DSM 33459 showed a unique metabolic profile as a producer of agmatine, a metabolite associated with metabolic health [18]. *Akkermansia* sp. DSM 33459 was able to improve body weight, liver weight, and total fat accumulation in a diet-induced obesity (DIO) mouse model. Significant improvements in serum insulin, the Homeostatic Model Assessment for Insulin Resistance (HOMA-IR) index, and resistin levels were also observed in the DIO mice. This report suggests that *Akkermansia* sp. DSM 33459 has novel properties as compared to the type strain and may be explored as a next-generation probiotic for metabolic health.

## 2. Materials and Methods

### 2.1. Isolation and Growth

The *Akkermansia* sp. DSM 33459 was isolated from a fecal sample collected from a healthy human subject. The strain was isolated on YCFA medium containing 10 g/L mucin. The fecal sample dilutions were plated onto YCFA + mucin agar plates. Plates were incubated at 37 °C in anaerobic boxes for a minimum of 72 h. Single colonies were picked and inoculated into 1 mL deep well plates with YCFA + mucin (10 g/L) for growth. Growth plates were incubated at 37 °C under anaerobic conditions for ~170 h. Aliquots were taken for 16S PCR and the remaining culture was frozen by the addition of glycerol to give a final glycerol concentration of 25%. Informed consent was obtained from the participant prior to participation. The clinical trial in which the sample was collected was approved by the Regional Ethics Committee of the Expert Responsibility area of Tampere University Hospital, Finland on 14 January 2016 (approval code: 115190).

### 2.2. 16S Genotyping, WGS and Phylogenetic Analysis 

The total DNA from the isolate was extracted using the PowerMag Microbial Extraction Kit from MoBio (now Qiagen, Hilden, Germany). Sequencing libraries were prepared with the Nextera Flex kit (Illumina, San Diego, CA, USA) and sequenced on MiSeq (Illumina, San Diego, CA, USA) in paired read 2 × 150 nt. The genome sequencing data were assembled using an in-house pipeline. In brief, reads were filtered and trimmed based on quality score then corrected using BFC [19]. The corrected reads were assembled using the SPAdes assembler [20]. The assembled contigs were corrected with Pilon [21]. After assembly, the Opening Reading Frames (ORFs) were predicted and annotated by Prokka [22]. Barrnap predicted 16S rRNA genes and the closest species identification by RDP pairwise alignment tool [23]. The multiple sequence alignment of 16S rRNA genes was performed using CLUSTALW [24]. Phylogenetic trees were reconstructed using the neighbor-joining method with 1000 bootstrap replications. The genome-wide average nucleotide identity (gANI) values between genomes were computed using OrthoANI [25] to assist species assignment based on whole genome data. 

To assess the similarity of other genomes from genus *Akkermansia*, a total of 235 public *Akkermansia* genomes were downloaded from NCBI RefSeq genome database. The protein ORFs from these genomes and DSM 33459 were clustered at 95% aa identity. The similarity of two genomes was then measured based on the proportion of the shared protein ORFs between each pair of genomes.

### 2.3. Determination of Optimum pH, Temperature and Carbon Source Utilization 

#### 2.3.1. Determination of Optimal pH for Growth

The media used for the determination of optimal pH for growth were brain heart infusion broth (HiMedia, Thane, India) + 10 g/L glucose. The pH was adjusted so that the pH range was 3.5 to 9, at intervals of 0.5 pH units. The pH adjustments were performed with HCl or NaOH, and the final volume of all media were made the same with sterile water. Each medium was filter sterilized. Media were dispensed, 2 mL each into 4 wells of an extended well 48-well plate. This distribution allowed for quadruplicate samples in the same plate, and all pHs were also in the same plate. Plates with media were covered with a breathable film and placed into an anaerobic chamber for at least 48 h. The pH of the media without any adjustment was 7.6.

Cells were obtained from cultures started in 25 mL sterile serum vials, in the same media at pH 7.6. Sealed serum vials were incubated at 37 °C with shaking at 240 rpm for 48 h. The OD_600_ of the cultures was measured to determine the volume to add to each well, so that the initial OD_600_ would be around 0.05 in each well. Once inoculated, test plates were covered with a breathable film, placed in anaerobic boxes with anaerobic sachets, and incubated at 37 °C with shaking to prevent cell clumping. Growth was determined by removing 100 µL aliquots at various times so that the ODs could be read on a plate reader. Samples were removed from the growth test plates from 0 to 214 h, with several sample time points in-between, for example 0 h, 52 h, 120 h, 166 h, and 214 h. There were at least quadruplicate readings for each sample at each time point. 

#### 2.3.2. Determination of Optimal Temperature for Growth

The media used to determine the optimal temperature for growth were either yeast casitone fatty acid (YCFA, Anaerobe System Inc., Morgan Hill, CA, USA) or BHI broth. Cell resuspensions were produced from previously streaked plates that had been growing for ~48 h under anaerobic conditions. The cells were removed from the agar plates using sterile inoculating loops and resuspended in 1–2 mL of media in a 50 mL sterile centrifuge tube. Cell clumps were broken apart by gentle pipetting and, additional volume of media was added, ~40 mL, and any remaining clumps were allowed to settle to the bottom of the tube. The OD_600_ was checked by taking an aliquot out of the chamber for measurement. The OD_600_ of the culture was adjusted by diluting with additional medium, to achieve the the desired OD_600_ of 0.05 to 0.075. For each strain, and each temperature being tested, 2 mL of cell culture was dispensed into 2 or 3 sterilized, labeled glass vials. The vials were sealed by adding a stopper plug and aluminum cap, and a crimp cap to keep the environment inside the vials anaerobic. The vials were then removed from the anaerobic chamber and incubated at designated temperatures. The temperature range tested was 25 to 55 °C at intervals of 5 °C. There were 2 or 3 vials of the same culture for each strain at each temperature tested. The vials were checked every day or every other day. After 48 to 72 h, a vial was removed from the incubator, opened, and the OD_600_ of the culture within was measured. After performing several of these experiments, it was determined that 48 h was a sufficient length of time to observe growth and the optimal growth temperature for both strains is 37 °C.

#### 2.3.3. Determination of Preferred Carbon Sources

The carbon sources were added to aliquots of YCFA media without glucose, each at a concentration of 40 g/L or 60 g/L. Media aliquots with the different carbon sources were dispensed to the wells of a 96-well standard microtiter plate. Media without added carbon were also added if needed, so that there were wells with carbon source concentrations of 20 g/L, 40 g/L, or 60 g/L. Glucose, glucosamine, galactosamine, galactose, fucose, and lactose were tested in the experiment.

Strains were previously streaked on YCFA + mucin at 10 g/L plates. Plates were then incubated for 48 h at 37 °C in an anaerobic box with sachets. Growth was collected from each plate using a sterile loop and resuspended in 0.5 mL YCFA media without glucose in a sterile microfuge tube. Cells clumps were broken up by gentle pipetting. An aliquot of the resuspended cells was diluted into a larger volume of the YCFA without glucose, enough so that there was at least 25 mL of culture with an OD_600_ of around 0.2. 

One hundred µL of these diluted cells from both strains was added to half the plate each, and gently mixed. The microtiter plate was covered with an ultra-clear seal. The microtiter plate was then transferred to the plate reader in the anaerobic chamber, and OD_600_ readings were taken every 15 min for around 48 h. All work was performed in an anaerobic chamber using a mixed gas of N_2_/CO_2_/H_2_ (85/10/5%), unless otherwise stated.

#### 2.3.4. Determination of Antibiotic Susceptibility

The antimicrobial susceptibility against ampicillin, vancomycin, gentamycin, kanamycin, streptomycin, erythromycin, clindamycin, tetracycline, chloramphenicol, ciprofloxacin, colistin, penicillin, benzylpenicillin, imipenem, meropenem, metronidazole, trimethoprim, and ampicillin-sulbactam was analyzed by Biosafe—Biological Safety Solutions Ltd., 70210 Kuopio, Finland using the broth microdilution method according to the Global Laboratory Standards Institute (CLSI) standard M11 (9th ed., 2018) [26], which describes the general method and the selection of medium and incubation conditions, except that custom-made microdilution trays were used (see below). The antimicrobial susceptibility against fosfomycin was analyzed using the agar dilution method according to the CLSI standard M11 (9th ed., 2018) [26] on Reinforced medium for Clostridia agar (RMFC). The OD_600_ was adjusted to 0.125 (±0.01), which corresponds to approximately 8.6 × 10^6^ CFU/mL based on a preliminary study. The cell suspension was further diluted in broth to reach 5 × 10^4^ CFU/well for the broth microdilution method or 10^5^ CFU/spot for the agar dilution method.

The test strain did not grow in standard growth medium (Brucella broth or agar supplemented with 5 μg/mL hemin, 1 μg/mL vitamin K, and 5% laked horse blood). Hence, the test was performed with custom-made Sensititre™ FINBIOS1 96-well plates (Thermo Fisher Scientific, Waltham, MA, USA) under anaerobic conditions using YCFAC broth, (Anaerobe Systems Inc., Morgan Hill, CA, USA), and with reinforced medium for Clostridia agar (Lab M, Heywood, UK) prepared in-house at 35 ± 2 °C for 48 ± 2 h. *Bacteroides fragilis* DSM 2151 (ATCC 25285) was used as a quality control strain using both standard broth and YCFAC broth.

The antibiotic susceptibility of *Akkermansia* sp. DSM 33459 was evaluated by comparing the minimum inhibitory concentrations (MICs) to the breakpoint values for Enterobacteriaceae defined by the European Food Safety Authority (EFSA) [27] and the breakpoint tables published for Gram negative anaerobes by the European Committee on Antimicrobial Susceptibility Testing (EUCAST) [28], as there are no established breakpoints for *Akkermansia* species. Additionally, the genome sequence was screened for the presence of antibiotic-related resistance genes, virulence factors, and toxins based on amino acid searches against the Comprehensive Antibiotic Resistance Database (CARD) [29] and the Virulence Factors Database (VFDB) [30], respectively, using BLAST [31].

### 2.4. FAME Analysis 

The analysis of cellular fatty acids was performed by Microbial ID Inc. (Newark, DE, USA), where strain DSM 33459 and type strain BAA835 underwent the standard sample preparation where they were grown on BHIA to extract the fatty acid methyl esters for identification. Samples were then loaded onto the gas chromatograph for analysis. Using the Sherlock^®^ pattern recognition software, a sample FAME profile was generated [32].

### 2.5. Metabolite Profiling

For comparison of metabolic capabilities of *Akkermansia* sp. DSM 33459 and *A. muciniphila* ATCC BAA-835, strains were grown in YCFAC media. The YCFAC media were inoculated with 1% overnight culture and supernatants were collected after 24 h of growth. Cells were separated by centrifugation at 8900× *g* for 5 min and then filtered by 0.2 um filter. Samples of media only were used as a control.

Cell-free supernatants were harvested and analyzed by CE-TOF-MS (Human Metabolome Technologies America, Inc., Boston, MA, USA). Cationic conditions were as follows: Samples were injected using 50 mbar, 10 s onto Fused Silica Capillary (i.d. 50 µm × 100 cm) on an Agilent CE-TOF System (Agilent Technologies Inc., Santa Clara, CA, USA). Cationic Buffer solution (1 M Formic acid) was used and a CE-voltage of 30 kV. Positive mode mass spectrometer conditions were as follows: MS Capillary voltage: 4.0 V, ESI Positive ionization mode, *m*/*z* range 50–1000. Anionic conditions were as follows: Samples were injected using 50 mbar, 22 s onto Fused Silica Capillary (i.d. 50 µm × 100 cm) on an Agilent CE-TOF System (Agilent Technologies Inc., Boston, MA, USA). Anionic Buffer solution (50 mM Ammonium acetate pH 7.5) was used and a CE-voltage of 30 kV. Negative mode mass spectrometer conditions were as follows: MS Capillary voltage: 3.5 V, ESI Negative ionization mode, *m*/*z* range 50–1000.

### 2.6. ATP Measurements 

YCFAC media with mucin (10 g/L) were inoculated with 1% overnight culture and spiked with 1 mM ATP. Supernatants were collected just after inoculation and after 8 h of growth. ATP levels were measured by using the ATP determination kit (A22066, Invitrogen, Waltham, MA, USA). Cells were separated by centrifugation at 8900× *g* for 5 min and then filtered by 0.2 µm filter. Cell-free supernatants were harvested and analyzed for the ATP levels as per the manufacturer’s instructions.

### 2.7. Culture and Pasteurization of Akkermansia *sp.* DSM 33459

*Akkermansia* sp. DSM 33459 was cultured anaerobically in a basal mucin-free medium. Cultures were washed and concentrated in anaerobic phosphate buffer saline (PBS) with 25% (*v*/*v*) glycerol under strict anaerobic conditions. Additionally, an identical quantity of *Akkermansia* sp. DSM 33459 was grown under similar conditions and was inactivated by pasteurization. *Akkermansia* sp. DSM 33459 was pasteurized for 30 min at 70 °C as described previously [15]. Cultures were then immediately frozen and stored at −80 °C. A representative glycerol stock was thawed under anaerobic conditions to determine the CFU/mL by flow cytometer (Apogee 40, Apogee Flow Systems Ltd., Hemel Hempstead, UK). Before administration by oral gavage, glycerol stocks were thawed on ice and diluted with anaerobic PBS to an end concentration of 10^9^ CFU/100 μL and 10% glycerol. Lyophilized *Akkermansia* sp. DSM 33459 was resuspended in anaerobically prepared PBS and then gavaged to the mice at a concentration equivalent to 10^9^ CFU/100 μL.

### 2.8. Animal Assays 

A total of 30 male nine-week-old C57Bl/6J DIO and 10 male C57Bl/6J mice were included in the study (10 animals per group) purchased from Jackson Laboratories (Bar Harbor, ME, USA). Following a two-week acclimation period, all animals were randomly allocated to three treatment groups of 10 individuals each, based on body weight. *Akkermansia* sp. DSM 33459 prepared as a suspension in PBS + glycerol (Akk^Gly^) or lyophilized (Akk^Lyo^) or pasteurized (Akk^Pas^) were administered daily (10^9^ CFU) by oral gavage for 12 weeks. Liraglutide (Advanced ChemBlocks Inc., Hayward, CA, USA) was used as a positive control and was administered at 0.2 mg/kg.

Upon arrival at the Testing Facility all DIO animals were maintained on high fat diet (Research Diets #D12492). Ten animals from the control group were maintained on standard rodent chow (PMI Nutrition International Certified Rodent Chow No. 5CR4). Test articles *Akkermansia* sp. (frozen, pasteurized, and lyophilized) were prepared daily and administered within 1 h of formulation. The dose volume for each animal was 100 uL. 

CR Institutional Animal Care and Use Committee (IACUC) approval for this study (Protocol 20255864) was received on 27 July 2020.

#### 2.8.1. Body Composition Measurements by EchoMRI™-500 A100

The animals were taken from their home cage and placed into a plastic restrainer compatible with the body composition analysis instrument. The restrainer was placed into the EchoMRI^TM^ system, body composition was assessed, and the animal was then returned to its home cage. Animals were in the restrainer for no longer than 5 min.

#### 2.8.2. 16S rRNA Amplicon Fecal Microbiota Profiling 

The 16S rRNA profiling was performed on mouse fecal pellet samples collected from Days −1, 8, 29, 57 and 79. Using sterile forceps, mouse fecal pellets were placed into the wells of a Bead Plate from the DNeasy PowerSoil HTP 96 Kit (Qiagen, Hilden, Germany), and DNA were extracted by using the manufacturer’s protocol. The variable V4 region (515F/806R primers) was amplified by PCR under the following conditions: 10 min at 95 °C followed by 35 cycles of 95 °C 15 s, 55 °C 30 s, and 72 °C for 2 min. PCR products were purified using AMPure XP beads (Beckman Coulter, Indianapolis, IN, USA) and Nextera XT indices (Illumina, San Diego, CA, USA) were added during a second PCR. Samples were pooled, quantified using the PicoGreen DS DNA Assay (Thermo Fisher Scientific, Waltham, MA, USA) and sequenced for 2 × 250 cycles on the MiSeq (Illumina, San Diego, CA, USA) per the manufacturer’s instructions. The resulting paired-end reads were merged using FLASH [33], primers were removed, and reads with an overall quality score less than 20 were discarded by the RDP Initial Process tool [23]. The reads were clustered at 99% similarity by CD-HIT [34] and those with abundance >0.1% were assigned taxonomy against the RDP 16S rRNA taxonomy training set no. 18 [35].

#### 2.8.3. Insulin and Resistin Measurements 

Serum resistin was measured by using the Mouse Adipokine Magnetic Bead Panel Kit (Millipore, Burlington, MA, USA, cat#: MADKMAG-71K) according to the manufacturer’s instructions. Fasting blood glucose and plasma insulin levels were measured using the Insulin Mouse Insulin ELISA Kit (ALPCO, Salem, NH, USA, cat# 80-INSMSU-E01). Insulin resistance index was calculated by using the homeostasis model assessment (HOMA) index. HOMA-IR was calculated according to the formula: fasting insulin (microU/L) × fasting glucose (nmol/L)/22.5.

### 2.9. Proteomics 

#### 2.9.1. Tissue and Sample Handling

Approximately 50 mg mouse liver was homogenized using beat beading on a Percellys 24 tissue homogenizer (Bertin Technologies SAS, Montigny-le-Bretonneux, France). Homogenization was achieved using a SDS lysis buffer (5% SDS in 100 mM triethylammonium bicarbonate, TEAB, pH 7.5). Homogenates were heated to 95 °C for 10 min to fully denature the proteins. DNA was sheered using a Bioruptor Pico (Diagenode Diagnostics, Liège, Belgium) high energy water bath sonicator system. Homogenates were centrifuged for 10 min at 14,000× *g* and supernatants were used for further analysis. Protein concentration of lysates was estimated using a Pierce™ Detergent Compatible Bradford Assay Kit (Thermo Fisher Scientific, Waltham, MA, USA). A volume equal to 50 µg protein was digested to peptides using a trypsin to protein ratio 1:25 and a 96-well S-trap assay (ProtiFi, Farmingdale, NY, USA) [36]. Peptide concentrations were measured with a Pierce Quantitative Colorimetric Peptide Assay (Thermo Fisher Scientific, Waltham, MA, USA) and adjusted accordingly to the same concentration.

#### 2.9.2. Data Acquisition 

Nano LC-MS/MS analyses were performed using an UltiMate™ 3000 RSLCnano System (Thermo Fisher Scientific, Waltham, MA, USA) interfaced to a Q Exactive™ HF Hybrid Quadrupole-Orbitrap™ Mass Spectrometer (Thermo Fisher Scientific, Waltham, MA, USA). Samples were dissolved in 0.1% TFA and loaded onto a 20 mm nanoViper Trap Column (Acclaim™ PepMap™ 100 C18, 3 µm particle size and an i.d. 0.075 mm) connected to a 600 mm analytical column (CoAnn integrated, 1.7 µm C18 beads and an i.d. 0.075 mm). Separation was performed with a flow rate of 250 nL/min using a 100 min gradient of 2–41% Solvent B (100% ACN, 0.1% FA) into the Nanospray Flex Ion Source (Thermo Fisher Scientific, Waltham, MA, USA). The Q Exactive HF instrument was operated in a data-dependent MS/MS using HCD fragmentation. The peptide masses were measured by the Orbitrap (MS scans were obtained with a resolution of 60,000 (FWHM) at *m*/*z* 200). The top 12 most intense ions were selected and subjected to fragmentation. Ions were isolated by the quadrupole using a 1.2 Da width isolation window. Fragment spectra were recorded in the Orbitrap with a resolution 30,000 (FWHM) at *m*/*z* 200. Dynamic exclusion was enabled with an exclusion duration of 16 s. Samples were analyzed in a randomized order with a QC sample after every fifth sample and with five QC samples in the beginning of the sample sequence. 

#### 2.9.3. Data Analysis 

The LC-MS/MS data was processed (smoothing, background subtraction, and centroiding) using Proteome Discoverer (Version 2.2, Thermo Fisher Scientific, Waltham, MA, USA). The processed LC-MS/MS data was submitted to database searching against a UniProt Rattus Norvegicus database containing 29,944 sequences using an in-house Mascot server. Trypsin was chosen as the enzyme with a maximum of two missed cleavages allowed. S-Carbamidomethyl cysteine was defined as a fixed modification and oxidation of methionine as variable modifications. The MS/MS results were searched with a peptide ion mass tolerance of ±10 ppm and a fragment ion mass tolerance of ±0.8 Da. Percolator [37] was used for calculating false discovery rates. Only peptides that were identified as rank 1 peptides and with a confidence value of 1% (q < 0.01) were considered for further analysis. 

All raw files were imported into Expressionist v.12.0.9 (Genedata, Basel, Switzerland) for data analysis. Imported files were noise filtered using chemical noise subtraction. Data were RT aligned using a pairwise alignment, filtered, and smoothed before peak detection, based on volumes. Detected peaks were isotopic clustered and singletons were filtered out. Peak clusters were matched with Proteome Discoverer identifications and peptides were grouped based on protein identifications. Proteins were quantified based on the three most intense peptides Hi-3 [38]. Quantitative results were exported into Analyst v.12.0.9 (Genedata, Basel, Switzerland) for normalization, statistical filtering, and testing. Principle Component Analysis was performed using the Genedata Analyst module. A total of 4240 proteins were identified and quantified. Data were normalized using an intensity drift normalization, where intensities from QC samples are used to correct for drifting in the LC-MS/MS performance.

### 2.10. Statistical Analysis

Data from body weight changes, resistin, and insulin measurements were analyzed by one-way ANOVA repeated measures followed by Dunnett’s multiple comparisons. For all analyses, the significance threshold was placed at *, *p* < 0.05; **, *p* < 0.01; ***, *p* < 0.001. In the figures, data are represented as mean ± SEM. Graphs were prepared by GraphPad Prism (Graphpad, San Diego, CA, USA).

#### Analysis for Proteomics

Proteomics was mapped by 4438 manifestations deriving from a cross-sectional study design, with 4 treatments (diet) randomly allocated to 39 subjects (mice) with 9–10 mice per group. The 4 treatments were (Chow, Vehicle, Akk^Gly^, Liraglutide) and the focus was on the 2 pairwise contrasts (Vehicle vs. Akk^Gly^) and (Vehicle vs. Liraglutide).

A multivariate latent class model was adopted [39]. Latent class and latent transition analysis was fitted to the data using PROC GLIMMIX from the statistical software SAS 9.4 (SAS Institute Inc., Cary, NC, USA, 2016). From the latent structure, the 2 pairwise contrasts were quantified in Z-scale and significance at level 5% was according to a 2-sided test (e.g., an absolute Z-score > 1.96).

## 3. Results

### 3.1. Strain Isolation and Characterization

The *Akkermansia* sp. DSM 33459 was isolated from a fecal sample collected from a healthy human subject. The samples selected for this isolation were shown to have a higher abundance of the strain *Akkermansia* sp. DSM 33459, based on the previously analyzed 16S community analysis. The isolation method used in this round was selection on YCFA medium containing mucin at 10 g/L.

#### Phylogenetic Characterization

The draft genome of strain *Akkermansia* sp. DSM 33459 is comprised of 31 contigs with N50 of 331,405 bp and 125× coverage. The genome size is 3.19 Mb, which is larger than the genome sizes of type strains of the other two *Akkermansia* species: *A. muciniphila* MucT (2.66 Mbp) and *Akkermansia glycaniphila* PytT (3.07 Mb). The G + C content of the genomic DNA is 57.7%. The 16S rRNA sequence of strain DSM 33459 is 1509 bases in length and showed 99.2% identity to 16S rRNA of the closest strain *A. muciniphila* ATCC BAA-835 and 95.4% identity to *A. glycaniphila* PytT. Since previously only two species were known in genus *Akkermansia*, a phylogenetic tree was reconstructed with these three strains and some strains from class Verrucomicrobiae that were included in the publication describing *A. glycaniphila* [2]. The phylogenetic analysis indicates that *Akkermansia* sp DSM 33459 is a member of the genus *Akkermansia*, with *A. muciniphila* MucT being its closest relative (Figure 1). The gANI between *Akkermansia* sp. DSM 33459 and *A. muciniphila* ATCC BAA-835 is 87.58%, and only 70.17% between *Akkermansia* sp. DSM 33459 and *A. glycaniphila* PytT. Based on the gANI values below the species boundary cutoff of 95% [40,41], strain DSM 33459 is proposed as a novel species within genus *Akkermansia*.

### 3.2. Microbial Characterization

#### 3.2.1. Growth Properties of *Akkermansia* sp. DSM 33459

*Akkermansia* sp. DSM 33459 was strictly anaerobic, nonmotile, and non-spore-forming. Its growth was characterized and compared with the type strain BAA-835. Both strains showed a very similar profile for pH and temperature optima (Table 1). The *Akkermansia* sp. DSM33459 strain showed differential growth on glucosamine and galactosamine as compared to BAA-835. The type strain BAA-835 showed a preference for glucose as a carbon source (Figure 2). *Akkermansia* sp. DSM 33459 showed pH optima of 6.5 as compared to 7.0 for the type strain. The type strains were able to grow between 25 °C and 40 °C, whereas *Akkermansia* sp. DSM 33459 showed a slightly narrower range between 30 °C and 40 °C. 

#### 3.2.2. Antibiotic Resistance and Safety Profile of *Akkermansia* sp. DSM 33459

The MIC values for tetracycline, colistin, and fosfomycin against *Akkermansia* sp. DSM 33459 were equal to or below the EFSA MIC cutoff values for Enterobacteriaceae (Table 2). The MIC values for ampicillin-sulbactam, clindamycin, chloramphenicol, meropenem, and metronidazole against *Akkermansia* sp. DSM 33459 were below the EUCAST MIC breakpoint values for Gram-negative anaerobes. The MIC values for ampicillin, gentamicin, kanamycin, streptomycin, and ciprofloxacin exceeded at least four-fold the EFSA MIC cut-off values for Enterobacteriaceae. The MIC values for benzylpenicillin and imipenem exceeded the EUCAST MIC breakpoint values for Gram-negative anaerobes. No MIC cut-off or breakpoint values are available for penicillin, vancomycin, erythromycin, or trimethoprim against *Akkermansia* sp. Even under the best controlled conditions, the isolate’s inherent biological variability and other factors may lead to a range of MIC values similar to what is observed with replicate testing with QC strains. The MIC values of ampicillin, penicillin, imipenem, and meropenem against the quality control strain *B. fragilis* DSM 2151 were significantly elevated when testing was performed in YCFAC broth instead of standard medium, suggesting that the test medium may have an effect on imipenem and meropenem tolerances (Appendix A). 

In silico analysis of the *Akkermansia* sp. DSM 33459 genome did not produce any significant matches to antibiotic resistance genes in the CARD database. Upon further manual inspection, a putative Class A beta-lactamase (EC 3.5.2.6) was identified in the genome, as well as a chromosomal point mutation in the indigenous *gyrA* gene that is a known active drug-binding site for ciprofloxacin. A search for decarboxylase genes encoding biogenic amines of human safety concern determined that these were absent from the genome, indicating *Akkermansia* sp. DSM 33459 cannot produce histamine, tyramine, putrescine, or cadaverine. Similar to the type strain ATCC BAA-835, carboxynorspermidine decarboxylase (EC 4.1.1.96) and carboxynorspermidine synthase (EC 1.5.1.43) enzymes were located in the genome, which indicates that *Akkermansia* sp. DSM 33459 may be capable of producing spermidine. There were no known virulence factors, hemolysin genes, bacteriocins, or toxins identified in the genome using an amino acid sequence similarity threshold of 80% that would present a safety concern for eukaryotic cells.

#### 3.2.3. Fatty Acid Analysis 

A sample comparison of the FAME profile showed a significant difference between *Akkermansia* sp. DSM 33459 and type strain BAA-835 (Table 3). A wide variety of cell wall fatty acids were detected in *Akkermansia* sp. DSM 33459. The FAME analysis showed that 15:0 was the predominant fatty acid in *Akkermansia* sp. DSM 33459. The type strain had a higher signature for 16:0 and 18:0 fatty acids. The type strain showed relatively less diversity in the fatty acid profile as compared to the DSM 33459 strain. The difference in the fatty acid profile may impact the way different strains interact with the host [42]. Lipids form the most important component of the bacterial membranes, and the unique set of lipids present on the cell membrane can modify the host innate immune response and can define the strain-to-strain variation in terms of their impact on host physiology. It will be interesting to further evaluate the impact of cell wall fatty acids on interactions with the host.

### 3.3. Identification of Cbi Operon in the Novel Akkermansia Strain 

A total of 235 public *Akkermansia* genomes were downloaded from the NCBI RefSeq genome database. The protein sequences from these genomes and DSM 33459 were clustered at 95% aa identity. The percentages of clusters not shared between each pair of genomes were then calculated to estimate the distance between them. The closer the genomes are, the fewer the non-shared clusters. DSM 33459 forms a tight cluster with these 16 public genomes (Appendix A). All these 16 genomes have gANI values >95% when compared to DSM 33459. In addition, the Cbi operons are present in 14 of these 16 genomes (Table 4). The cbi operons are also present in three genomes: GCF_002884975.1, GCF_902387265.1, and GCF_002884915.1. These three genomes did not cluster with DSM 33459, but they formed a cluster that was the closest to the DSM 33459 cluster. The type strain lacks cbi operon (Appendix A). Genome pairs with a gANI value of >95% typically belong to the same species. These results indicate that most of the strains from this DSM 33459 cluster have Cbi operons, and all of the strains with cbi operons belong to this DSM 33459 cluster or the closest neighboring cluster.

### 3.4. Agmatine Production by Akkermansia *sp.* DSM 33459 

To explore the metabolic capabilities of *Akkermansia* sp., supernatants were collected after 24 h of growth and analyzed by CE-TOF-MS and compared with the supernatants collected from the type strain. The supernatants from *Akkermansia* sp. DSM 33459 showed an accumulation of agmatine as compared to the media control. Agmatine was not detected in the supernatants collected from the type strain (Figure 3). The metabolite Agmatine (N-(4-aminobutyl)guanidine) was detected in cationic mode at retention time 4.23 min at *m*/*z* 131.130 m.u, and the identification was performed against a known standard. 

#### Genome Analysis Showed Presence of Agmatine Synthesis Pathway

Functional pathway mining conducted using the MetaCyc Pathway Tools feature [43] indicated that there is an arginine decarboxylase enzyme (EC 4.1.1.19) present in the genome of *Akkermansia* sp. DSM 33459. This enzyme, which is encoded by the *speA* gene, facilitates the conversion of arginine to agmatine and the release of CO_2_ (Figure 4). A search for a similar protein conducted in the genome of *A. muciniphila* ATCC BAA-835 revealed that the type strain contains a non-identical homolog (Amuc_1658) with a pairwise nucleotide identity of 91.5% and an aa similarity of 98.5%. 

### 3.5. Removal of Extracellular ATP by Akkermansia *sp.* DSM 33459

It has been shown that *A. muciniphila* can negate inflammation in the gut but the mechanism pertaining to this effect is not well understood. Extracellular ATP has been shown to induce inflammation [44]. Extracellular ATP acts via the P2Y family of receptors [45]. We evaluated the strains *Akkermansia* sp. DSM 33459 and *A. muciniphila* BAA-835 for their capabilities to remove ATP from the growth media. *Akkermansia* sp. was able to remove ATP more efficiently as compared to *A. muciniphila* ATCC BAA-835 (Figure 5) and possibly can negate the inflammation mediated by extracellular ATP. This observation provides a possible mode of action for the beneficial effects of *Akkermansia* against gut inflammation. 

#### Genome Analysis Showed Presence of ATPase

The depletion of extracellular ATP in culture medium has been studied in various Gram-negative bacteria; however, the specific mechanism of transport and hydrolyzation is not well understood [46,47,48]. Similar to what has been reported in *Escherichia coli* and *Salmonella enterica*, an alkaline phosphatase (EC 3.1.3.1) annotated as phoA and a 5′-nucleotidase (EC 3.1.3.5) annotated as ushA were located in the genomes of both *Akkermansia* sp. DSM 33459 and *A. muciniphila* ATCC BAA 835, indicating both strains have the potential to hydrolyze ATP to adenosine in the periplasmic space (Figure 5) [46,47,48]. A search for these enzymes in publicly available *Akkermansia* genomes revealed that both of these enzymes and adjacent genomic regions appear relatively conserved in Akkermansia species (data not shown). Despite the capability for the strains to hydrolyze ATP, the mechanism for the transport of ATP across the inner membrane remains elusive. An ATP transporter or uptake system has not been reported in bacteria, and, similarly, we did not find genomic evidence of an ATP translocase in either strain [46]. Alternatively, it is also possible that hydrolyzation of ATP may instead be occurring on the cell surface [47].

### 3.6. Akkermansia *sp.* DSM 33459 Regulates the Diet-Induced Obesity Associated Metabolic Markers

*A. muciniphila* has attracted attention due to its association with host health. The relative abundance of *A. muciniphila* is inversely correlated with obesity in humans [49] and it was shown to correct insulin resistance and obesity while increasing gut barrier function in a mouse model of diet-induced obesity [14]. These observations make *A. muciniphila* an attractive probiotic candidate. Until this date all the data from pre-clinical or clinical work have been concentrated on *A. muciniphila.* In the current study, the novel strain of *Akkermansia* (DSM 33459) was evaluated in a DIO model to determine its impact on metabolic markers. The *Akkermansia* sp. DSM 33459 prepared as a suspension in PBS + glycerol (Akk^Gly^) or lyophilized (Akk^Lyo^) or pasteurized (Akk^Pas^) were administered daily (10^9^ CFU) by oral gavage for 12 weeks. Mice on a high fat diet (HFD) showed a significant increase in body weight as compared to mice on normal chow. *Akkermansia* sp. DSM 33459 body weight gain was significantly reduced in the Akk^Gly^ group compared to the vehicle control in DIO mouse model. The Akk^Gly^ group also showed a significant improvement in total fat weight and liver weight (Figure 6A–C). The resistin and insulin levels were measured in serum and the Akk^Gly^ group showed improved resistin insulin levels in the serum (Figure 7 and Figure 8). Akk^Lyo^ administration also improved resistin levels (Figure 7). Liraglutide was able to improve all the endpoints as compared to the vehicle control (Figure 6 and Figure 7).

Plasma insulin levels were significantly elevated in the HFD group mice (Figure 8). Based on these measurements, we calculated the insulin resistance index by using the homeostasis model assessment (HOMA) index. The HOMA-IR index was found to be significantly lower in the normal, Akk^Gly^, and Liraglutide groups compared to the HFD control (Figure 9). This result suggests that Akk^Gly^ can play an important role in reducing the development of obesity-induced insulin resistance in mice.

#### 3.6.1. *Akkermansia* sp. DSM 33459 Engrafts in Colon

Engraftment was studied by analyzing the DNA extracted from the mice fecal pellets. Mice on HFD showed a significant increase in native *A. muciniphila* abundance when compared to the group on a normal diet. Interestingly, *Akkermansia* sp. DSM 33459 was able to replace this native population efficiently in the Akk^Gly^ group. *Akkermansia* sp. DSM 33459 showed significant presence in the samples from days 8 to 79. The engraftment was not as strong in the Akk^Lyo^ group. As expected, pasteurized *Akkermansia* did not show any engraftment (Figure 10).

#### 3.6.2. *Akkermansia* sp. DSM 33459 Modulates Several Liver Pathways

From the proteomics analysis, we found relative abundances of 4240 liver proteins. Using a multivariate LCA model, 62 upregulated and 229 down regulated proteins were identified when the Akk^Gly^ group was compared to the vehicle alone. The liraglutide-treated mice showed 972 upregulated and 509 down regulated proteins. We noted that several peroxisomal proteins had a similar direction of regulation between the Akk^Gly^ and the Liraglutide drug control groups (Figure 11 and Appendix A). Among these regulated proteins are Pex5, Pex6, and Pex19, which have been shown to play a critical role in peroxisome function [50].

## 4. Discussion

In this study, we described the identification and characterization of a new species of *Akkermansia*: *Akkermansia* sp. DSM 33459, isolated from human feces of a healthy donor. The bacterium *A. muciniphila* represent 3–5% of the microbial composition in the healthy human GI tract [51]. It has been shown that *A. muciniphila* is associated with better health and that it is depleted in subjects with metabolic disorders [52]. One of the most prominent associations in mice and human studies is a negative correlation with obesity, insulin, and glucose management [53]. The administration of live or pasteurized *A. muciniphila* was shown to negate the risk of metabolic disorders by improving glucose and insulin levels in both mice and humans [14,16,54]. One of the first reports to show the association of *Akkermansia* with metabolic health was published by Zhang et al., in 2009, where they showed that Verrucomicrobia phylotypes related to an *Akkermansia* sp. (GenBank accession: AJ400275) were prominent in the normal weight group and in the post-gastric bypass group but were poorly represented in the obese group [55].

In the last decade there has been a significant increase in our understanding in the role of *A. muciniphila* in human health. *Akkermansia* abundance varies with age and geographical location and changes with health status. The Analysis of American Gut Project has shown that *Akkermansia* abundance is associated with a low risk of obesity and its abundance declines with age [52]. The role of *Akkermansia* has been extended from metabolic impact to inflammation, immune modulation, and impact on neurological disorders [11,56].

There is a significant diversity among human isolates of *A. muciniphila* [4]. The evolution of diversity in *Akkermansia* suggests a possible role of diet and host environment. It is certainly possible that some strains may be more beneficial than others depending upon the nature of their interaction with the host. In addition, differences in metabolic capabilities of strains may be critical for their efficacy. Short chain fatty acid production and interaction of *Akkermansia* membrane protein (Amuc1100) with gut epithelial cells have been proposed as possible mechanisms [57]. 

*Akkermansia* sp. DSM 33459 has 87.5% identity with the type strain *A. muciniphila* BAA-835 based on gANI. *Akkermansia* sp. DSM 33459 was tested side by side with the type strain to evaluate the impact of genomic differences. *Akkermansia* sp. DSM 33459 showed significant differences in terms of preferred carbon sources where the DSM 33459 strain preferred glucosamine and galactosamine, and glucose was the preferred carbon source for the BAA-835 strain. Fatty acid analysis showed that *Akkermansia* sp. DSM 33459 has a very different composition compared to the type strain BAA-835. Genomic analysis showed that *Akkermansia* sp. DSM 33459 harbors a cbi operon that is absent in BAA-835 strain. The cbi operon is involved in the synthesis of the corrin ring of vitamin B_12_ (cobalamin). Vitamin B_12_ plays an essential role in gut physiology [58]. Cobalamin is an unusual vitamin in that it is not made by plants and instead is synthesized exclusively by bacteria and archaea [59]. Due to its essential nature and limited availability in the human gut, vitamin B_12_ is regarded as a key element in host-microbe interactions [58]. In a recent study, Kirmiz et al., analyzed 75 *Akkermansia* strains and reported that they differ in their potential to produce this important cofactor based on genome analysis and bioassays [6]. Bacteria produce a variety of cobalamins and the nature of the cobalamin produced by *Akkermansia* is not known. Thus, it is difficult to predict its impact on host physiology. However, it is feasible that the vitamin B_12_ produced by DSM 33459 may have an impact on the quality and the quantity of the metabolites produced by the bacterium itself, as it has been shown that B_12_ is critical for the production of propionate [6]. Propionate production was detected in the *Akkermansia* sp. DSM 33459 supernatant. There is a possibility that the vitamin B_12_ produced by *Akkermansia* sp. DSM 33459 may promote metabolic activities in other commensals. The ability to synthesize vitamin B_12_ broadens the potential range of metabolic benefits provided by *Akkermansia* sp. DSM 33459.

Metabolic analysis of the bacterial growth supernatants has shown production of agmatine by *Akkermansia* sp. DSM 33459. Agmatine has been demonstrated to have an impact on human health, ranging from diabetes to neurodegenerative diseases. Possible molecular targets of agmatine have been described, including neurotransmitter receptors, ion channels, transporters, and advanced glycation end product (AGE) formation [60]. Agmatine is biosynthesized from arginine by the enzyme arginine decarboxylase (ADC). It has been demonstrated that microbial-derived agmatine plays an important role in metabolic health. Bacterial agmatine regulates host metabolism and lifespan in *Caenorhabditis elegans* [61]. By conducting a host-microbe-drug-nutrient screen, agmatine was identified as a key bacteria-derived effector of metformin therapy [61].

The impact of *Akkermansia* sp. DSM 33459 on metabolic health was evaluated in a DIO mouse model. No significant variation was observed in food intake but significant improvements in body weight were documented. *Akkermansia* sp. DSM 33459 showed 10% less body weight as compared to the control mice. This effect was not observed in the same way in the groups where mice received *Akkermansia* in a lyophilized or pasteurized format. Liver weight showed a similar trend, whereby Akk^Gly^ was able to reduce the liver weight significantly as compared to the vehicle control. Consistent with this reduced body weight gain, treatment with *Akkermansia* sp. DSM 33459 was associated with lower fat mass. Body weight composition was determined by using NMR [62]. The body composition was measured at days −1 and −2 and days 83 and 84. *Akkermansia* sp. DSM 33459 was able to reduce body fat accumulation significantly as compared to the control mice. Liver tissues were subjected to untargeted proteomics and analysis showed that the group that received liraglutide altered the expression of several proteins involved in peroxisome biogenesis and β-oxidation. *Akkermansia* was also able to alter the expression of several proteins involved in the peroxisome pathway, a very similar effect to the drug control. 

Resistin plays a critical role in metabolic health by modulating various pathways. Resistin is expressed and secreted by adipose tissue in mice and plays an important role as a mediator in obesity-induced insulin resistance [63,64]. The impact of *Akkermansia* sp. on fat mass was consistent with the changes in the circulating levels of the adipokine resistin. The vehicle control on HFD showed an elevation in circulating resistin levels compared to the mice on a chow diet, which was significantly reduced by *Akkermansia* sp. DSM 33459 treatment. It has been shown previously that treatment with pasteurized *A. muciniphila* resulted in a much lower insulin concentration when compared to the HFD group, resulting in a lower insulin resistance (IR) index [15]. Contrary to this report, we found that mice treated with *Akkermansia* showed a lower insulin concentration during the study as compared to the pasteurized *Akkermansia*. These differences between *Akkermansia* sp. DSM 33459 and *A. muciniphila* suggest either a different mode of action or a different susceptibility of the active molecule to the pasteurization process.

It has been shown that *A. muciniphila* has anti-inflammatory properties, but the mode of action is not well defined. The metabolic potential of *Akkermansia* sp. DSM 33459 to catabolize pro-inflammatory metabolites was tested and it was observed that *Akkermansia* sp. DSM 33459 can metabolize ATP in the media more efficiently compared to the type strain. Extracellular ATP released by immune cells and by microbiota in the intestinal lumen induce a variety of immune responses that mediate intestinal homeostasis via P2 purinergic receptors [65]. Accumulation of ATP signaling leads to mucosal immune system disruption, which leads to pathogenesis of intestinal inflammation. The metabolic ability of *Akkermansia* sp. DSM 33459 to remove ATP from its surroundings suggests *Akkermansia* can negate inflammation by modulating the purigenic signaling. Recently, there has been significant advancement in understandings of the impact of ExATP on gut physiology [65,66]. The ability of *Akkermansia* sp. DSM 33459 to metabolize ATP presents an interesting and unexplored beneficial impact of this bacterium on health.

It is noteworthy that *Akkermanisa* sp. DSM 33459 was able to colonize the mouse GI tract. Interestingly, DSM 33459 was able to replace the native mouse strain *A. muciniphila*. It has been proposed that vitamin B_12_-producing phylotypes may potentially outcompete other phylogroups when levels of vitamin B_12_ precursors in the GI tract are scarce [4]. Colonization was less effective in the group that received *Akkermania* sp. DSM 33459 in a lyophilized form, suggesting that colonization is important for the efficacy in this model. Reduced colonization may also provide a possible explanation for the reduced efficacy of the pasteurized format. DSM 33459 exhibits some unique metabolic capabilities as compared to the type strain. DSM 33459 showed production of agmatine and vitamin B_12_, and the ability to remove ATP; if these properties contribute to its efficacy, then this provides an explanation for the lower response in the Akk^Pas^ group. Contrary to our expectations, Akk^Lyo^ did not show comparable efficacy to Akk^Gly^, and this may be attributable to a less efficient reconstitution of the samples prior to administration, in comparison to the *Akkermansia* prepared from frozen glycerol stocks. In this experiment, the lyophilized format was resuspended in PBS and gavaged. The strain may need a longer period for hydration and possible activation to demonstrate efficacy. In mice, the transit time is between 3–4 h and this may not be sufficient for the strain to become metabolically and transcriptionally active and execute its full efficacy.

We evaluated the antibiotic resistance profile of *Akkermania* sp. DSM 33459, as probiotics should not contribute to the spread of antibiotic resistance. The breakpoint cut-offs from the EFSA FEEDAP panel are for Enterobacteriaceae, so there are inherent differences expected in the antibiotic susceptibility profiles between facultative and obligate Gram-negative bacteria. The MICs were above the recommended breakpoints for the aminoglycosides, for which obligate anaerobes are intrinsically resistant [67] as well as ampicillin, ciprofloxacin, and benzylpenicillin. The MIC value for imipenem only exceeded the EUCAST MIC breakpoint value for Gram-negative anaerobes by one dilution step, which is considered normal variation around the mean and does not indicate resistance. A putative Class A beta-lactamase (EC 3.5.2.6) gene was annotated in the genome, which may confer some low-level resistance to penicillins and carbapenems; however, this strain is sensitive to penicillins with the addition of a beta-lactamase inhibitor, which is commonly used in clinical settings. We found no genomic evidence that any putative resistance genes would be a risk for mobilization to another organism. Additionally, we did not find evidence of any known virulence factors, hemolysin genes, toxins, or biogenic amines of concern in the genome; therefore, we report no apparent concerns regarding the safety and antibiotic resistance profiles of *Akkermansia* sp. DSM 33459 for its potential use as a probiotic.

*Akkermansia* has been at the center of several interesting studies. Despite the 16S and gANI variations, *Akkermansia* strains have been considered a single species with a focus on mucin degradation. The strain differences at the functional level are largely unexplored. Here, a significantly different strain of *Akkermansia* belonging to a new *Akkermansia* species with unique genomic and metabolic signatures was characterized. In addition, the antibiotic resistance profile, genomic safety, and efficacy of the strain in resolving metabolic disorders in a diet-induced obesity mouse model were evaluated, and these results suggest that *Akkermansia* sp. DSM 33459 presents a novel candidate for a next-generation probiotic for improving human health.

## Figures and Tables

**Figure 1 cells-11-02084-f001:**
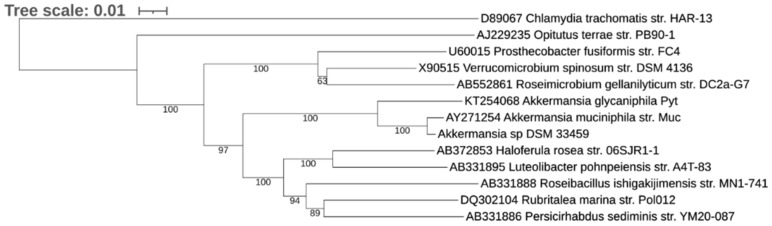
A phylogenetic tree consisting of *Akkermansia* sp. DSM 33459. The tree was reconstructed with the neighbor-joining method with 1000 bootstraps. Numbers represent bootstrap values. The legend bar indicates 5% sequence divergence. *Chlamydia trachomatis* was used as an outgroup.

**Figure 2 cells-11-02084-f002:**
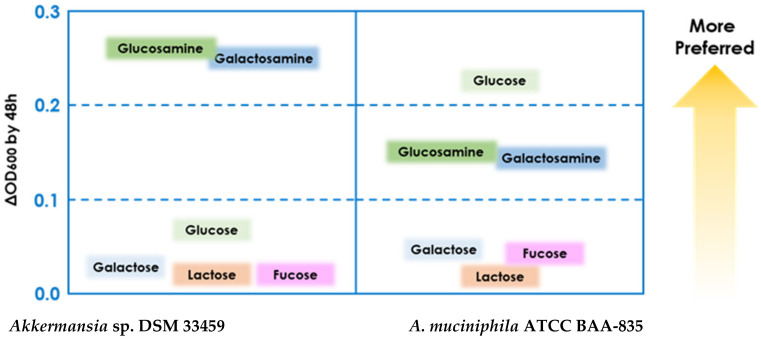
Determination of preferred carbon source for growth.

**Figure 3 cells-11-02084-f003:**
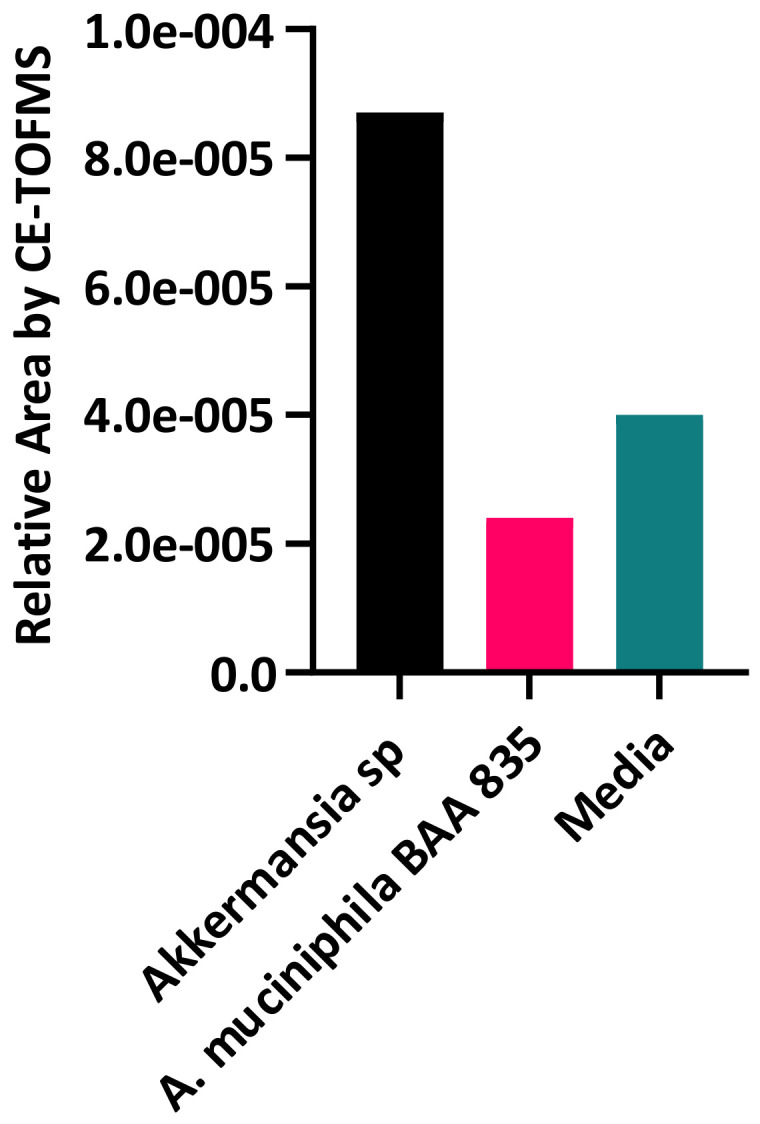
Sample comparison of CE-TOFMS relative peak areas of Agmatine. *Akkermansia* sp. DSM 33459 and *A. muciniphila* BAA-835 supernatants were collected after 24 h of growth and analyzed by CE-TOF-MS.

**Figure 4 cells-11-02084-f004:**
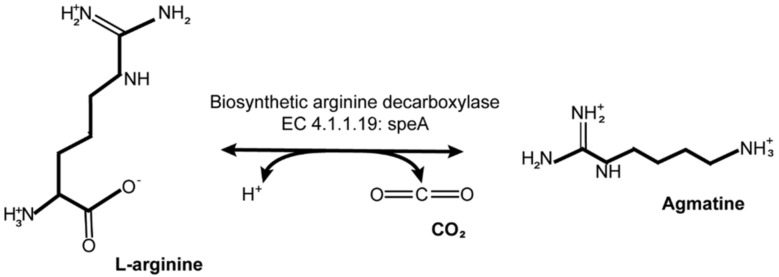
MetaCyc pathway for bacterial conversion of arginine to agmatine (ID: PWY0−1299) [43].

**Figure 5 cells-11-02084-f005:**
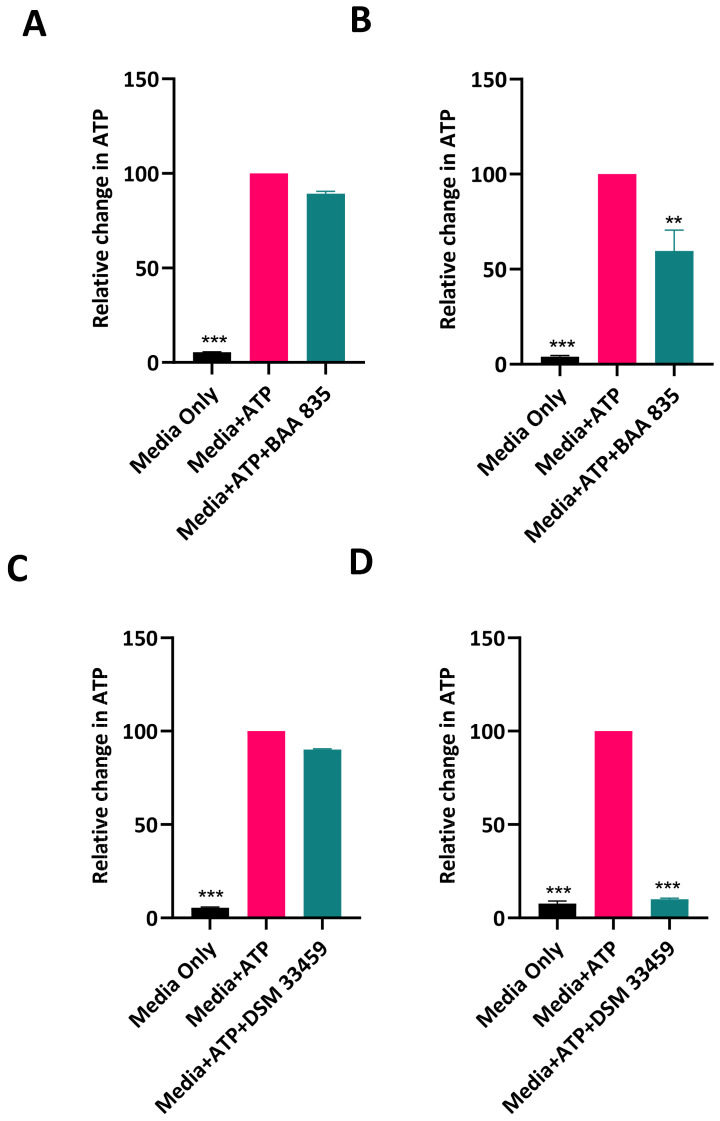
ATP amounts were detected in the supernatants for *A. muciniphila* ATCC BAA-835 at time 0 (**A**) and after 8 h of growth (**B**). ATP was estimated in the supernatants from *Akkermansia* sp. DSM 33459 at time 0 (**C**) and after 8 h of growth (**D**). Each experiment was conducted with duplicate wells and was repeated at least three times. Data are presented as mean ± SEM. One-way, two tailed ANOVA followed by Dunnett’s multiple comparisons test were used for statistical analysis. **, *p* < 0.01; ***, *p* < 0.001, compared to Media + ATP.

**Figure 6 cells-11-02084-f006:**
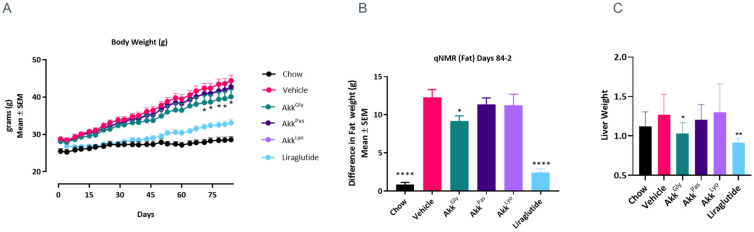
*Akkermansia* sp. DSM33449 negates weight increase and reduces fat accumulation in a DIO mouse model (*n* = 10). (**A**) Growth curve of animals fed a standard chow (Chow) or high-fat diet (HFD) and treated by oral gavage with a solution of vehicle (PBS + Glycerol) or *Akkermansia* sp. DSM 33459 frozen (Akk^Gly^) or pasteurized (Akk^Pas^) or lyophilized (Akk^Lyo^) for 12 weeks. (**B**) Difference in fat accumulation over the 12-week treatment period. (**C**) Difference in liver weight over the 12-week treatment period. Data are presented as mean ± SEM. *, *p* < 0.05; **, *p* < 0.01; ****, *p* < 0.0001, compared to vehicle.

**Figure 7 cells-11-02084-f007:**
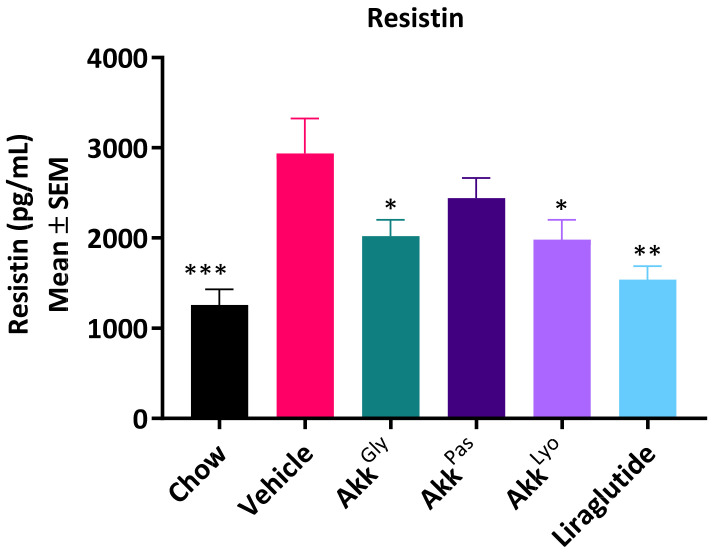
*Akkermansia* sp. DSM33449 modulates resistin levels in a DIO mouse model (*n* = 10). Serum resistin levels of animals fed on a standard chow (chow) or high-fat diet (HFD) and treated by oral gavage with a solution of vehicle (PBS + Glycerol) or *Akkermansia* sp. DSM 33459 frozen (Akk^Gly^) or pasteurized (Akk^Pas^) or lyophilized (Akk^Lyo^) for 12 weeks. Data are presented as mean ± SEM. *, *p* < 0.05; **, *p* < 0.01; ***, *p* < 0.001, compared to vehicle.

**Figure 8 cells-11-02084-f008:**
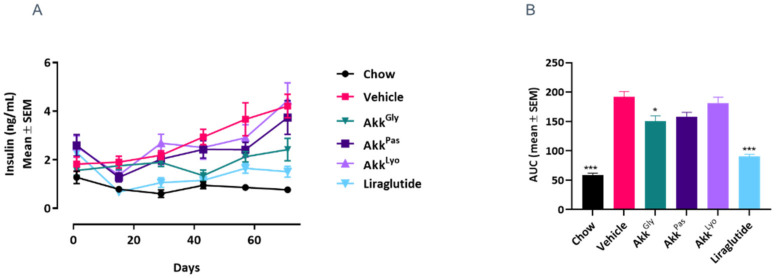
*Akkermansia* sp. DSM33449 modulates the insulin levels in a DIO mouse model (*n* = 10). (**A**) Serum insulin levels of animals fed on a standard chow (Chow) or high-fat diet (HFD) and treated by oral gavage with a solution of vehicle (PBS + Glycerol) or *Akkermansia* sp. DSM 33459 frozen (Akk^Gly^) or pasteurized (Akk^Pas^) or lyophilized (Akk^Lyo^) for 12 weeks. (**B**) AUC for serum insulin levels. Data are presented as mean ± SEM. *, *p* < 0.05; ***, *p* < 0.001, compared to vehicle.

**Figure 9 cells-11-02084-f009:**
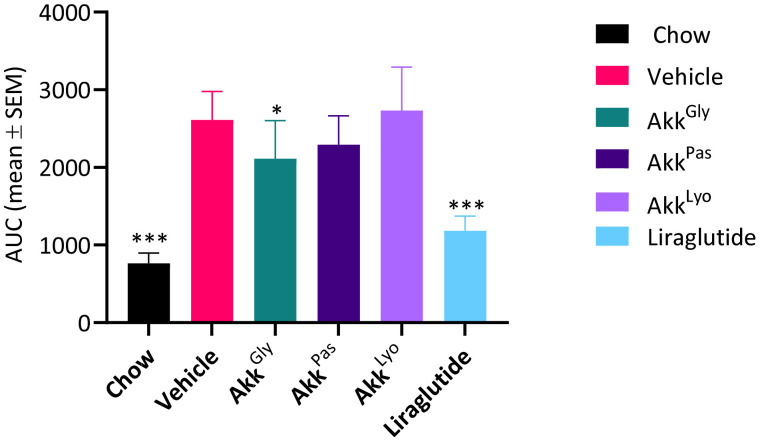
*Akkermansia* sp. DSM33459 improves HOMA-IR in a DIO mouse model (*n* = 10). Serum insulin and glucose levels were measured and used for the determination of HOMA-IR. HOMA-IR of animals fed on a standard chow (Chow) or high-fat diet (HFD) and treated by oral gavage with a solution of vehicle (PBS + Glycerol) or *Akkermansia* sp. DSM 33459 frozen (Akk^Gly^) or pasteurized (Akk^Pas^) or lyophilized (Akk^Lyo^) for 12 weeks. Data are presented as mean ± SEM. *, *p* < 0.05; ***, *p* < 0.001, compared to vehicle.

**Figure 10 cells-11-02084-f010:**
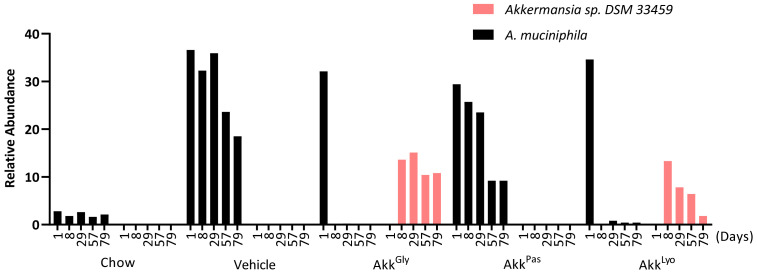
Engraftment of *Akkermansia* sp. DSM 33459 in the mice GI tract. Relative abundance of *A. muciniphila* and *Akkermansia* sp. DSM 33459 in mice fecal samples at different time points.

**Figure 11 cells-11-02084-f011:**
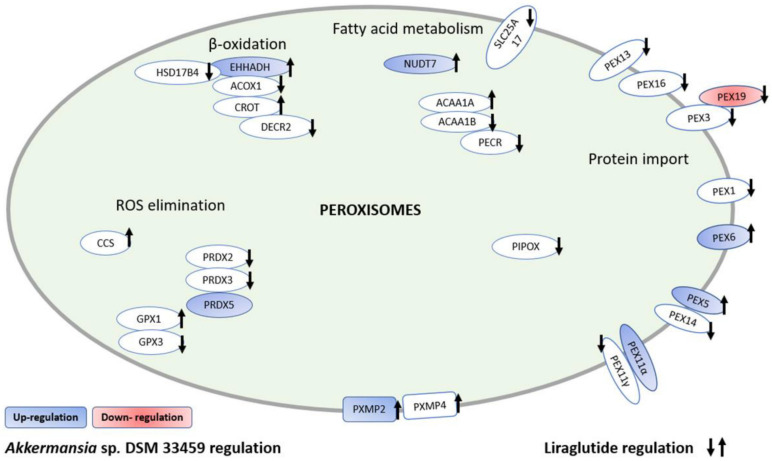
Similar modulation of peroxisomal proteins by *Akkermansia* sp. DSM 33459 and drug control group.

**Table 1 cells-11-02084-t001:** Determination of Optimal Temperature for growth.

	*Akkermansia* sp. DSM 33459	*A. muciniphila* ATCC BAA-835
25 °C	−	+
30 °C	+	+
35 °C	+	+
37 °C	+	+
40 °C	+	+
45 °C	−	−
50 °C	−	−

**Table 2 cells-11-02084-t002:** Susceptibility of *Akkermansia* sp. DSM 33459 to selected antimicrobials using YCFAC broth or RMFC agar (fosfomycin).

Antimicrobial	MIC (µg/mL)	EFSA MIC Cut-off Values for *Enterobacteriaceae* (µg/mL)	EUCAST MIC Breakpoint Values for Gram-Negative Anaerobes (µg/mL)
**Ampicillin**	>32	8	2
**Ampicillin-sulbactam**	1/0.5 ^a^		8/4
**Benzylpenicillin**	>8		0.5
**Penicillin**	>16		
**Gentamicin**	32	2	
**Kanamycin**	512	8	
**Streptomycin**	>512	16	
**Clindamycin**	0.12		4
**Tetracycline**	2 ^b^	8	
**Ciprofloxacin**	128 ^c^	0.06	
**Colistin**	≤0.5	2	
**Fosfomycin ***	8	8	
**Vancomycin**	8		
**Erythromycin**	16		
**Chloramphenicol**	2 ^d^		8
**Imipenem**	8		4
**Meropenem**	4		8
**Metronidazole**	0.5		4
**Trimethoprim**	4		

The values are the mode of three replicate analyses (values that differ from the mode are noted in the footnote). ^a^ One of the three replicates presented an MIC of 0.5/0.25 μg/mL. ^b^ One of the three replicates presented an MIC of 4 μg/mL. ^c^ One of the three replicates presented an MIC of 64 μg/mL. ^d^ One of the three replicates presented an MIC of 4 μg/mL. * Reinforced medium for Clostridia agar was used.

**Table 3 cells-11-02084-t003:** Comparative Chemotaxonomic characteristics of the strains *Akkermansia* sp. DSM 33459 and *A. muciniphila* BAA-835.

Fatty Acid	DSM 33459	ATCC BAA-835
**10:00**	0.11	
**13:00**	0.4	
**14:00 iso**	6.27	2.32
**14:00**	0.44	2
**14:00 DMA**	1.92	
**15:0 iso**	1.65	7.55
**15:00 antiso**	51.62	29.73
**15:00**	9.71	0.69
**15:0 3OH**	8.44	0.9
**16:0 iso**	3.75	4.08
**16:1 w7c**	0.18	
**16:00**	2.19	20.38
**16:0 OH**	0.45	
**17:0 iso**	0.3	0.98
**17:0 anteiso**	2.25	7.84
**17:00**	3.11	
**17:0 anteiso 3OH**	0.56	
**17:0 3OH**	2.08	
**18:1 w9c**	0.51	18.18
**18:00**	0.49	1.2
**un 18.199**		0.7
**18:1 w7c DMA**		0.67
*** Summed Feature 5**	2.38	
**Summed Feature 6**	0.55	
**Summed Feature 9**	0.47	
**Summed Feature 10**		1.95
**Summed Feature 11**	0.18	
**Summed Feature 12**		0.84

* Summed features represent groups of two or more fatty acids that are grouped together for the purpose of evaluation of the MIDI Sherlock^®^ Microbial Identification System. Fatty acid methyl ester contents are expressed in percentage abundance.

**Table 4 cells-11-02084-t004:** Genome comparison based on Corrin ring biosynthesis gene cluster.

Genome Accession	Presence of Cbi Operon	Cluster with *Akkermansia* sp. DSM 33459	*A. muciniphila* DSM 33459 gANI%
GCF_002885015.1	yes	yes	99.94
GCF_002885555.1	yes	yes	99.91
GCF_017566105.1	yes	yes	98.84
GCF_010231335.1	yes	yes	98.81
GCF_018336995.1	yes	yes	98.79
GCF_010230375.1	yes	yes	98.79
GCF_002885535.1	yes	yes	98.71
GCF_002885055.1	yes	yes	98.69
GCF_002885105.1	no	yes	98.63
GCF_002885075.1	yes	yes	98.61
GCF_002885025.1	yes	yes	98.59
GCF_002885095.1	yes	yes	98.56
GCF_902387295.1	yes	yes	98.56
GCF_002885135.1	no	yes	98.41
GCF_002885515.1	yes	yes	98.18
GCF_002884995.1	yes	yes	98.17
GCF_002884975.1	yes	no	90.84
GCF_902387265.1	yes	no	90.84
GCF_002884915.1	yes	no	90.73

## Data Availability

Not applicable.

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
