# Peer review of "Identification and Characterization of a Novel Species of Genus Akkermansia with Metabolic Health Effects in a Diet-Induced Obesity Mouse Model"

_cells, 2022, doi:10.3390/cells11132084_

Round 1

Reviewer 1 Report

I would like to thank authors for the interestig material. I have just some proposals, which could make it more involving for scientists.

First, Akkermansia isolation method should be described in more detail. 

Then, due to the described differences I would suggest to make an accent on the safety of this novel strain.

And some minor remarks:

"Fatty Acid Analysis" part could be discussed in more detail.

16s rRNA analisys of mice feaces could also could be more widely represented.

In general, the article seems to be a holistic work with a high-quality design and in-depth analysis. Really nice and elegant paper.

Reviewer 2 Report

The submission “Identification and characterization of a novel species of genus Akkermansia with metabolic health effects in a diet induced obesity mouse model” describes the isolation and comprehensive characterization of a novel strain of Akkermansia with interesting properties to be used as a probiotic. The authors describe step-by-step the isolation, and culturing, biochemistry, and genomic characterizations of this interesting Akkermansia sp. DSM 334 strain.

As minor remarks:

Table 1 appears not to be necessary since data could be mentioned in the results section.

Table 3 title is too long.

Figure 3, this image cannot be properly appreciated by the reader. It may go to supplementary files.

Figures 7-10, the font size of legends should be increased for clarity.

Figure 11, Authors should provide a more friendly graphic representation of these important data results to the readers.

Reviewer 3 Report

Kumar et al studied the differences between Akkermansia sp DSM 33459 with A. muciniphila ATCC BAA-835 in terms of fatty acids profile and carbon source. The authors also evaluated the influence of Akkermansia sp DSM33459 on obesity in mice. The manuscript is well done, however, some points should be improved. 

Material and methods

It is not clear why the authors used in some experiments BHI and another YCFA medium to grow Akkermansia. Can you explain why you used BHI for pH determination and optimal temperature using YCFA or BHI?

Akkermansia is described as oxygen-sensitive bacteria. However, it is not clear if the authors perform the experiments using an Anaeboric chamber for all experiments. Can you explain why the inoculum was shaken at 240 rpm (line 121)? Please, clarify it. 

 In section 2.3.3., the authors should indicate what is the carbon sources. 

During experiments, the authors used different OD to adjust the concentration of Akkermansia. Please, could clarify why using different OD in different assays and the concentration of viable bacteria should be written. 

Results

The legend of Table 2 should be improved.  What is the meaning of + or -? Presence or absence?

Figure 2. The results did not explain if the medium was supplemented with 20, 40, or 60 g/L of carbon source  (as mentioned in section 2.3.3.). Moreover, the results should be expressed in terms of the number of viable bacteria by CFU counting, because OD measures total bacteria (live and dead bacteria). Therefore, an increase in OD did not mean that Akkermansia are alive. 

Table 3 should be improved. The authors should explain the empty boxes. In the discussion section is not clear if EFSA and EUCAST MIC values were evaluated by microdilution or agar dilution methods. Section 2.3.4 is not also clear on the method used for MIC determination. 

Table 4, should be improved. There are empty boxes, why? Is there not this lipid or the amount is lower than the limit of detection of equipment? 

To simplify the reading, the table should have the common name of lipid, lipid numbers, and structural formula. The values obtained are expressed in terms of concentration, retention time, and % fatty acid composition? This information should be included. 

Figure 3 is impossible to analyze. The quality of the figure must be improved.

Minor comments

Line 62. "other references?"

You should revise the manuscript, there are double spaces between sentences. 

 OD600 or A600? please revise it

Please revise the units mL, h, min, and scientific notation. 

Line 212. "filtered by 0.22 uM", amend to 0.22um

Line 212 and 230 "centrifugation at 10 000 rpm", should be indicated the rotor or expressed in g

Reviewer 4 Report

Introduction. The presented results of the study are relevant. The manuscript is interesting on the problem of the symbiotic microorganisms. The introduction has the main objectives and key concepts of this manuscript. The results are bestowed in a logical order.

Materials and Methods. Used up-to-date techniques. Metabolite profiling: Indicate for readers what was used as a control (this information is available, but it is in the results. It must be included in the materials and methods). The authors performed sequencing of the studied strain. Are these data included in the NSBI or other public databases (if so, indicate the ID).

References.  It is necessary to correct No. 24 in the list of references (twice the year is indicated).

Round 2

Reviewer 3 Report

The authors improved the manuscript by attending to the comments of reviewers. However, the improvement of the Tables was not significant and evident.